# Transcription Factor *Sm*SPL2 Inhibits the Accumulation of Salvianolic Acid B and Influences Root Architecture

**DOI:** 10.3390/ijms232113549

**Published:** 2022-11-04

**Authors:** Xiangzeng Wang, Yao Cao, Jiaxin Yang, Tong Zhang, Qianqian Yang, Yanhua Zhang, Donghao Wang, Xiaoyan Cao

**Affiliations:** Key Laboratory of the Ministry of Education for Medicinal Resources and Natural Pharmaceutical Chemistry, National Engineering Laboratory for Resource Development of Endangered Crude Drugs in Northwest of China, Shaanxi Normal University, Xi’an 710062, China

**Keywords:** auxin, cytokinin, root architecture, salvianolic acid B, *Salvia miltiorrhiza*, SmSPL2

## Abstract

The SQUAMOSA PROMOTER-BINDING PROTEIN-LIKE (SPL) transcription factor play vital roles in plant growth and development. Although 15 SPL family genes have been recognized in the model medical plant *Salvia miltiorrhiza* Bunge, most of them have not been functionally characterized to date. Here, we performed a careful characterization of *SmSPL2*, which was expressed in almost all tissues of *S. miltiorrhiza* and had the highest transcriptional level in the calyx. Meanwhile, *Sm*SPL2 has strong transcriptional activation activity and resides in the nucleus. We obtained overexpression lines of *SmSPL2* and *rSmSPL2* (miR156-resistant *SmSPL2*). Morphological changes in roots, including longer length, fewer adventitious roots, decreased lateral root density, and increased fresh weight, were observed in all of these transgenic lines. Two *rSmSPL2*-overexpressed lines were subjected to transcriptome analysis. Overexpression of *rSmSPL2* changed root architectures by inhibiting biosynthesis and signal transduction of auxin, while triggering that of cytokinin. The salvianolic acid B (SalB) concentration was significantly decreased in *rSmSPL2*-overexpressed lines. Further analysis revealed that *Sm*SPL2 binds directly to the promoters of *Sm4CL9*, *SmTAT1*, and *SmPAL1* and inhibits their expression. In conclusion, *SmSPL2* is a potential gene that efficiently manipulate both root architecture and SalB concentration in *S. miltiorrhiza*.

## 1. Introduction

*Salvia miltiorrhiza* Bunge is a renowned model medical plant species [1]. Its dried root is a traditional Chinese medicine (referred to as Danshen in Chinese), which has been extensively utilized for the remedy of cardiovascular and cerebrovascular diseases [2]. Water-soluble salvianolic acid B (SalB) is one of the main bioactive constituents of Danshen and has a wide range of pharmacological activities, which include antioxidant, anti-inflammatory, antiapoptotic, antifibrotic effects; promoting stem cell proliferation and differentiation; and more [3]. The SalB content in Danshen has been listed as an indicator of quality control [4].

Recently, several strategies, such as treatment with an elicitor, manipulation of the key enzyme genes in the SalB biosynthesis pathway, and the application of transcription factors (TFs), have been employed to improve the availability of SalB in vivo [5]. Among these strategies, TFs are considered as an effective tool to control the biosynthesis of the main bioactive constituents in *S. miltiorrhiza* [6]. A few families of TFs, such as R2R3-MYBs (R2R3-type myeloblastosis proteins), bHLHs (basic helix-loop-helix protein), WKRYs, AP2/ERFs (APETALA2/ETHYLENE RESPONSE FACTORs), GRAS (named after the first three identified proteins within this family, GAI, RGA, and SCR), and LBDs (lateral organ boundary domain proteins), have been reported to regulate SalB biosynthesis and influence its accumulation [6].

Since the root of *S. miltiorrhiza* is the medicinal part, its morphology and biomass is an important agronomic trait. Although some agricultural production measures, such as proper planting densities and fertilization combinations, have been used to improve root production [7], genetic manipulation strategies have not been reported to date. For *S. miltiorrhiza* breeders, the ultimate goal is to acquire high-quality *S. miltiorrhiza* resources, which possess not only a high active ingredient content, but also high root production. Identifying potential genes, which participate in the regulation of root morphology and biomass, is the prerequisite of the molecular breeding of *S. miltiorrhiza*.

The SQUAMOSA PROMOTER-BINDING PROTEIN-LIKE (SPL) TFs are exclusive to plants and play vital roles in plant growth and development, such as controlling plant leaf development, nutritional phase transition, flowering time, fruit development, root development, anthocyanin accumulation, and stress responses [8]. Among the 16 *SPL* genes of *Arabidopsis thaliana*, ten are targeted by miR156 and clustered into five clades in accordance with the amino acid sequences of the conserved DNA binding domains–*AtSPL2/AtSPL10/AtSPL11*, *AtSPL9/AtSPL15*, *AtSPL3/AtSPL4/AtSPL5*, *AtSPL6*, and *AtSPL13* [9]. *AtSPL2* affects plant fertility and floral organ development due to the direct activation of *ASYMMETRIC LEAVES 2* [10]. *AtSPL3*, *AtSPL9*, and *AtSPL10* participate in the inhibition of lateral root development in *Arabidopsis*, with *AtSPL10* playing a predominant character [8]. *At*SPL9 interacts with *At*PAP1 (Production of Anthocyanin Pigment 1, a well-known R2R3-MYB member in Arabidopsis) to negatively regulate anthocyanin biosynthesis [11]. *At*SPL2, *At*SPL10, and *At*SPL11 bind directly to the promoters of *AP2/ERFs* and inhibit wound-induced auxin biosynthesis to suppress root regeneration [12].

For *S. miltiorrhiza*, 15 *SPL* genes have been found in its genome; eight of which, including *SmSPL2*/*3*/*5*/*6*/*8*/*11*/*14*/*15*, have been validated to be the targets of miR156 [13]. To date, only two *SPL* genes, *SmSLP6* and *SmSPL7*, have been functionally characterized in *S. miltiorrhiza* [14,15]. The overexpression of *SmSPL6* promotes the biosynthesis of phenolic acids and reduces the production of anthocyanin [14]. *Sm*SPL7 has been proved to be a positive regulator for anthocyanin accumulation while a negative regulator for phenolic acid in *S. miltiorrhiza* [15]. The functions of the other *SmSPL* genes have not been characterized to date.

*Sm*SPL2 and *At*SPL2/10/11 were clustered in the same subfamily based on phylogenetic tree analysis [13]. We speculated that *SmSPL2* may affect root growth and the production of bioactive constituents in *S. miltiorrhiza*. In the current work, we studied the function of *SmSPL2* by analyzing its expression profiles and gain-of-function phenotype. The overexpression of *SmSPL2* (*SmSPL2*-OE) and miR156-resistant *SmSPL2* (*rSmSPL2*-OE) in *S. miltiorrhiza* resulted in obvious changes in the adventitious root morphology, and the transcriptome of *rSmSPL2*-OE lines was further analyzed. *Sm*SPL2 was proved to be a negative regulator for SalB biosynthesis by directly suppressing the transcription of several enzyme genes for the pathway of phenolic acid biosynthesis. The present study confirmed that *Sm*SPL2 was a strong regulator for both SalB accumulation and root development in *S. miltiorrhiza*.

## 2. Results

### 2.1. SmSPL2 Expression Patterns

We detected the expression profiles of *SmSPL2* in different tissues of one-year-old *S. miltiorrhiza* by quantitative reverse transcription PCR (qRT-PCR) analysis. The qRT-PCR results revealed that *SmSPL2* was expressed in various tissues of *S. miltiorrhiza*, and the highest transcriptional level was found in the calyx, followed by the pistil (Figure 1A).

To further detect the temporal and spatial expression profiles of *SmSPL2*, we generated transgenic *Arabidopsis* plants in which the *GUS* gene was driven by *SmSPL2* promoter (*ProSmSPL2::GUS*). The histochemical GUS staining was performed on T2 generation transgenic *Arabidopsis* at various developmental periods. The result showed that the GUS signal was observed in both vegetative and reproductive stages of *Arabidopsis* (Figure 1). A strong GUS signal was visible in the whole seedlings after germination at two days (Figure 1B), five days (Figure 1C), and nine days (Figure 1D), with stronger signals in the older leaves than the younger leaves. During the anthesis stage of *Arabidopsis*, the GUS signal was observed in stem leaf (Figure 1E), rosette leaf (Figure 1F), stem (Figure 1G), flower (Figure 1H), and root (Figure 1I), with stronger signals in the tap root and calyx. The blue signal was obvious at both ends of the pod (Figure 1J). Our results suggested that the *SmSPL2* promoter drove the expression of genes in almost all plant components at various developmental periods.

### 2.2. SmSPL2 Resides in the Nucleus and Exhibits Transactivation Activity

To check the subcellular localization of *Sm*SPL2, we constructed recombinant vector *35Spro*::*SmSPL2*-*GFP*, which was instantaneously expressed in the onion epidermis. Our results revealed that the signal of green fluorescent GFP in the positive control was uniformly distributed throughout the cell, while that of the *Sm*SPL2-GFP fusion protein was specifically located in the nucleus (Figure 2A), which indicated that *Sm*SPL2 was nuclear protein.

Meanwhile, the recombinant pGBKT7-*SmSPL2* construct was used for the transactivation assay of *Sm*SPL2 in transformed yeast AH109. AH109 cells containing pGBKT7-*SmSPL2* or pGBKT7 plasmid grew normally on SD/-Trp solid medium; nevertheless, only cells containing pGBKT7-*SmSPL2* showed normal growth and turned blue on SD/-Trp/-Ade/-His/X-α-gal yeast medium (Figure 2B). These results revealed that *Sm*SPL2 had a transcriptional activation function.

### 2.3. Generation of SmSPL2/rSmSPL2-Overexpressed Transgenic S. miltiorrhiza

To analyze the roles of *SmSPL2* in *S. miltiorrhiza*, we obtained transgenic *S. miltiorrhiza* plants that overexpressed *SmSPL2*. Since *SmSPL2* is a target of miR156 [13], we also generated *rSmSPL2*-overexpressed (OE) *S. miltiorrhiza* plants. Compared with the control, the morphology of adventitious roots (AR) changed significantly in both *SmSPL2*-OE and *rSmSPL2*-OE lines, exhibiting longer root lengths and a reduced number of AR and lateral roots (Figure 3A–C). The qRT-PCR was used to analyze the transcriptional abundance of *SmSPL2* in the transgenic lines. Our results revealed that the transcriptional abundance of *SmSPL2* was dramatically increased in *SmSPL2*-OE and *rSmSPL2*-OE lines, in contrast to the control (Figure 3D). Among the transgenic lines, the expression levels of *SmSPL2* in *rSmSPL2*-OE1, *rSmSPL2*-OE2, and *rSmSPL2*-OE7 were 83.17-, 53.07-, and 42.66-fold of that of the control, respectively.

The phenotypic changes in the roots in the *SmSPL2*-OE1, *SmSPL2*-OE2, *SmSPL2*-OE4, *rSmSPL2*-OE1, *rSmSPL*-2OE*2*, and *rSmSPL2*-OE7 lines were obvious. Therefore, we measured AR number, AR length, and fresh weight of roots for the two-month-old transgenic lines. Compared with the control, the AR length was significantly increased in all these lines (Figure 3E), while the AR number was considerably decreased (Figure 3F). The fresh weight of the root was significantly increased in *SmSPL2*-OE1, *SmSPL2*-OE2, *SmSPL2*-OE4, and *rSmSPL2*-OE2 (Figure 3G). These results revealed that the overexpression of *SmSPL2* or *rSmSPL2* strongly affects the *S. miltiorrhiza* root phenotype.

### 2.4. Transcriptome Analysis of Overexpressed rSmSPL2 and S. miltiorrhiza Control Plants

To further investigate the potential molecular mechanisms of *SmSPL2* on the phenotype changes in *S. miltiorrhiza* plants, we selected *rSmSPL2*-OE2, *rSmSPL2*-OE7, and a control for RNA-seq. The Illumina deep sequencing obtained ~42.59, 41.59, and 44.06 million high-quality clean reads for *rSmSPL2*-OE2, *rSmSPL2*-OE7, and the control, respectively. The Q30 value (percentage of sequences with a sequencing error rate <0.1%) was 92.75% for *rSmSPL2*-OE2, 93.42% for *rSmSPL2*-OE7, and 93.31% for the control.

A grand total of 25,174 genes were recognized in the three cDNA libraries. We compared the global expression profiles of *rSmSPL2*-OE2 and *rSmSPL2*-OE7 versus the control. Transcriptome analysis identified 10,757 differentially expressed genes (DEGs) in *rSmSPL2*-OE2, with 5482 upregulated genes and 5272 downregulated genes (Appendix A). In *rSmSPL2*-OE7, 5511 DEGs (2455 upregulated genes and 3056 downregulated genes) were identified (Appendix A).

We listed 20 significantly enriched metabolic pathways for those DEGs by KEGG analysis (Appendix A). The *rSmSPL2*-OE2 and *rSmSPL2*-OE7 shared nine significantly enriched pathways, including phenylpropanoid biosynthesis, flavonoid biosynthesis, starch and sucrose metabolism, zeatin biosynthesis, carotenoid biosynthesis, and more (Appendix A). Among the significantly enriched metabolic pathways, phenylpropanoid biosynthesis was the most significant pathway in both *rSmSPL2*-OE2 and *rSmSPL2*-OE7, which indicated that *SmSPL2* was involved in phenylpropanoid biosynthesis.

### 2.5. DEGs Involved in Cytokinin Biosynthesis and the Signal Transduction Pathway

The cytokinin class of phytohormones has been implicated in various aspects of plant growth and development, including the inhibition of root growth and lateral root formation [16]. Since the DEGs in *rSmSPL2*-OE2 and *rSmSPL2*-OE7 were significantly enriched in the KEGG pathway of zeatin biosynthesis (Appendix A), we further analyzed all the DEGs involved in cytokinin biosynthesis and signaling pathway. As shown in Figure 4A,B, five DEGs encoding putative IPT (isopentenyltransferase), CYP735A, and LOG (LONELY GUY) for cytokinin biosynthesis and eight DEGs involved in cytokinin signal transduction were all up-regulated in *rSmSPL2*-OE2. For *rSmSPL2*-OE7, although two of the four DEGs encoding enzymes for cytokinin biosynthesis were downregulated, two DEGs involved in cytokinin signal transduction were up-regulated. Our transcriptome data indicated that the overexpression of *rSmSPL2* robustly triggered cytokinin signaling in *S. miltiorrhiza* roots.

Furthermore, we examined cytokinin levels in the roots of one-month-old *rSmSPL2*-OE2, *rSmSPL2*-OE7, and the control. Consistent with the transcriptome data, cytokinin levels in *rSmSPL2*-OE2 and *rSmSPL2*-OE7 were significantly upregulated, compared to the control (Figure 4C).

### 2.6. DEGs Involved in Auxin Biosynthesis and the Signal Transduction Pathway

It is well documented that the crosstalk between auxin and cytokinin signaling plays an important role in the regulation of root growth [17]. We further analyzed the DEGs involved in auxin biosynthesis and the signal transduction pathway (Figure 4D,E). Among the DEGs that encoded putative enzymes for auxin biosynthesis, including TAA (Tryptophan aminotransferase of *Arabidopsis*), AIM (Indole-3-acetamide hydrolase), and AAO (Aldehyde oxidase), one (*TAA1*) was down-regulated and two (*TAA2* and *AIM1*) were up-regulated in *rSmSPL2*-OE2, while two (*TAA1* and *AAO1*) were down-regulated, and one (*TAA2*) was up-regulated in *rSmSPL2*-OE7. The DEGs involved in auxin signal transduction, which encode the putative IAA (AUXIN (AUX)/INDOLE-3-ACETIC ACID) protein, GH3 (Gretchen Hagen 3), and SAUR (small auxin up RNA), were all down-regulated in *rSmSPL2*-OE2 (15 DEGs) and *rSmSPL2*-OE7 (five DEGs). The transcriptome data revealed that the overexpression of *rSmSPL2* strongly inhibited auxin signaling in the roots.

Additionally, we determined auxin levels in the roots of one-month-old *rSmSPL2*-OE2 and *rSmSPL2*-OE7. As shown in Figure 4F, the auxin concentrations in *rSmSPL2*-OE2 and *rSmSPL2*-OE7 were only 9.8% and 20.1% of the control, respectively, suggesting overexpression of *rSmSPL2* significantly suppressed auxin biosynthesis.

### 2.7. DEGs Involved in SalB Biosynthesis Pathway

To evaluate whether the decreased accumulation of SalB was the result of the altered transcription levels of genes in the salvianolic acid biosynthesis pathway, we examined all the possible enzyme genes (*SmTATs*, *SmHPPRs*, *SmPALs*, *SmC4Hs*, *Sm4CLs*, *SmRASs*, and *SmCYP98A14*) identified in the transcriptome data (Figure 5A). Among the 24 detected genes, 10 DEGs were found in *rSmSPL2*-OE2 with 9 being down-regulated (*SmTAT1*, *SmPAL1*, *SmPAL3*, *SmC4H1*, *Sm4CL1*, *Sm4CL3*, *Sm4CL10*, *SmRAS2*, and *SmRAS3*). For the six DEGs in *rSmSPL2*-OE7, five (*SmTAT1*, *SmPAL3*, *Sm4CL9*, *SmRAS2*, and *SmRAS3*) were down-regulated.

To inspect the accuracy of the transcriptome data, we selected six enzyme genes (*SmTAT1*, *SmPAL1*, *SmC4H1*, *Sm4CL1*, *SmCYP98A14*, and *SmRAS1*) for qRT-PCR analysis. Our qRT-PCR results revealed that all six tested genes were significantly suppressed in *rSmSPL2*-OE2 and *rSmSPL2*-OE7, which was relatively aligned with the RNA-seq data (Figure 5B). Our qRT-PCR results testified that the RNA-seq data was statistically reliable and the overexpression of *SmSPL2* suppressed the salvianolic acid biosynthesis pathway.

### 2.8. Overexpressed rSmSPL2 Inhibits RA and SalB Biosynthesis

Since SalB is synthesized via the pathways of tyrosine and phenylpropanoid metabolism [18], we speculated that *SmSPL2* may be involved in SalB biosynthesis, where rosmarinic acid (RA) is believed to be the SalB precursor [19]. We detected the RA and SalB in the roots of control, *rSmSPL2*-OE2, and *rSmSPL2*-OE7 via high-performance liquid chromatography (HPLC). As shown in Figure 5C, the retention time of RA and SalB was 31.37 ± 0.02 and 40.59 ± 0.02 min, respectively. Compared with the control, the RA and SalB contents significantly decreased in the *rSmSPL2*-OE transgenic lines. The RA concentrations in *rSmSPL2*-OE2 and *rSmSPL2*-OE7 were 59% and 78% of the control, respectively (Figure 5D). For SalB, the concentrations in *rSmSPL2*-OE2 and *rSmSPL2*-OE7 were only 20% and 36% of the control, respectively (Figure 5D). These results suggested that the overexpression of *rSmSPL2* suppressed RA and SalB biosynthesis in the roots of *S. miltiorrhiza*.

### 2.9. SmSPL2 Binds to and Represses Sm4CL9, SmTAT1, and SmPAL1 Promoters

SPL transcription factor can bind to the GTAC motif of target genes to regulate their expression [20]. We found that the promoter regions of quite a few genes in the salvianolic acid biosynthesis pathway (*Sm4CL9*, *SmTAT1*, *SmPAL1*, and *SmCYP98A14*) contain the GTAC motif (Figure 6A). To test whether *Sm*SPL2 directly binds to their promoters, yeast one-hybridization (Y1H) experiment was carried out. The Y1H results revealed that *Sm*SPL2 binds to the promoter regions of the three genes (*Sm4CL9*, *SmTAT1*, and *SmPAL1*) (Figure 6B).

Additionally, we performed dual luciferase transient transcriptional assays (TTA) in tobacco to investigate whether *SmSPL2* might inhibit their expression. The TTA results revealed that *Sm*SPL2 inhibited the expression of *Sm4CL9*, *SmTAT1*, and *SmPAL1* (Figure 6C,D). Overall, our results indicated that *Sm*SPL2 directly suppressed transcription of *Sm4CL9*, *SmTAT1*, and *SmPAL1* by binding to their promoters.

## 3. Discussion

### 3.1. Role of SmSPL2 in the Biosynthesis of Phenolic Acids

Because of the wide-ranging pharmacological effects of SalB, improving its production in vivo through biotechnology is one of the research hotspots for *S. miltiorrhiza* [21]. The most direct strategy is to change the expression levels of enzyme genes in the SalB biosynthesis pathway. For example, overexpressing *SmRAS* and *SmCYP98A14*, which encode the key enzymes for SalB biosynthesis, significantly increased the content of SalB in *S. miltiorrhiza* hairy roots [22]. Several TFs, such as *Sm*MYB97 [23], *Sm*MYB111 [24], *Sm*MYC2 [25], *Sm*ERF115 [26], *Sm*bZIP1 [27], and *Sm*NAC1 [28], positively regulate SalB biosynthesis by directly targeting the promoter regions of some key enzyme genes of phenolic acid biosynthesis and activating their expression in *S. miltiorrhiza*, while other TFs, such as *Sm*LBD50 [29], *Sm*bZIP2 [30], *Sm*GRAS1/2/3 [31], *Sm*ERF1L1 [32], *Sm*MYB36 [33], and *Sm*bHLH37 [34], have been verified to be the negative regulators for SalB biosynthesis.

Among the 16 SPL family members in *S. miltiorrhiza*, *Sm*SPL6 and *Sm*SPL7 have been experimentally proven to participate in the accumulation of SalB [14,15]. *Sm*SPL6 promotes SalB biosynthesis by directly activating the transcription of *SmCYP98A14* and *Sm4CL9* [14]. *Sm*SPL7 suppresses the biosynthesis of SalB by directly inhibiting the transcription of *SmTAT1* and *Sm4CL9* [15]. In the present study, we confirmed that *Sm*SPL2 was a negative regulator for SalB biosynthesis and that overexpression of *rSmSPL2* dramatically decreases the concentrations of SalB and RA in *S. miltiorrhiza* roots (Figure 5). Y1H and dual luciferase assays revealed that *Sm*SPL2 inhibited SalB biosynthesis by directly suppressing the transcription of multiple genes, including *SmTAT1*, *Sm4CL9*, and *SmPAL1* (Figure 6).

### 3.2. SmSPL2 Influences S. miltiorrhiza Root Architecture

In *Arabidopsis*, *At*SPL2, *At*SPL10, and *At*SPL11 inhibit root regeneration by impeding wound-induced auxin biosynthesis [12]. Phylogenetic analysis indicated that *Sm*SPL2 tended to cluster with *At*SPL2/10/11 [13]. We wondered whether *Sm*SPL2 was involved in root growth and development. In this study, obvious morphological changes in the roots, including longer length, fewer adventitious roots, decreased lateral root density, and increased fresh weight, were observed in *SmSPL2* or *rSmSPL2-*overexpressed transgenic *S. miltiorrhiza* (Figure 3). We used RNA-seq to investigate the likely molecular mechanisms that overexpressed *rSmSPL2* employed to influence root architectures. Transcriptome analysis showed that the DEGs in *rSmSPL2*-OE2 and *rSmSPL2*-OE7 were significantly enriched in the KEGG pathway of zeatin biosynthesis (Appendix A).

Cytokinin phytohormones have been known to inhibit root growth, since they were first discovered [35]. Cytokinin was observed to inhibit the adventitious roots formation near the cut end of cucumber hypocotyls [36]. The exogenous application of cytokinin inhibits the initiation of lateral roots and results in fewer lateral roots in Arabidopsis [37]. Mutants with increased cytokinin signaling have fewer lateral roots [38], whereas those with decreased cytokinin sensitivity have an increased number of lateral roots [39]. Transcriptome data indicated that cytokinin biosynthesis and the signal transduction pathway was significantly activated in transgenic lines (Figure 4), which is potentially one the most important reasons for morphological changes in the roots.

It is well documented that cytokinin and auxin interact antagonistically to regulate the size of the root apical meristem and the formation of lateral roots [40]. Auxin plays critical role in the control of the development and architectures of root systems and is a positive regulator of lateral root formation [41]. During lateral root development, auxin regulates cytokinin biosynthesis and signaling pathways, while cytokinin affects the pathways of auxin biosynthesis and transport [40]. Auxin biosynthesis, polar transport, and signal transduction are pivotal processes for lateral and adventitious root development [42]. Transcriptome analysis hinted that the overexpression of *rSmSPL2* strongly inhibited auxin signaling in the roots of transgenic lines (Figure 4). Additionally, the endogenous cytokinin levels in *rSmSPL2*-OE2 and *rSmSPL2*-OE7 was significantly increased, while the endogenous auxin levels were dramatically decreased when compared with the control. Taken together, we speculated that the morphological changes of the roots in transgenic lines were to a large extent attributed to the alteration of auxin and cytokinin biosynthesis and signaling.

## 4. Materials and Methods

### 4.1. Experimental Materials and Hormone Treatments

*S. miltiorrhiza* seeds were collected in the experimental field at Shaanxi Normal University. The seeds were sterilized, followed by germination on the MS medium to obtain sterile seedlings, which were used for genetic transformation, as previously described [43]. *S. miltiorrhiza*, *A. thaliana* ecotype Columbia-0, *Nicotiana tabacum* were planted under the following conditions: 22 °C under extended daylight conditions (16 h light: 8 h dark).

One-year-old *S. miltiorrhiza* plants growing in the experimental field at the flowering stage were collected and its different parts, including tap roots, lateral roots, stems, leaves, pistils, stamens, corollas, and calyxes, were separately frozen in liquid nitrogen for RNA extraction.

### 4.2. RNA Extraction and qRT-PCR

Total RNA was extracted using Tissue Total RNA Isolation Kit (Vazyme, Nanjing, China), followed by a reverse transcription using HiScript II Reverse Transcriptase (Vazyme, Nanjing, China). SYBR green qPCR Mix (Vazyme, Nanjing, China) was used for the qRT-PCR experiment on the LightCycler^®^ 96 real-time PCR assay system (Roche, Switzerland). *SmUbiquitin* served as the reference gene.

### 4.3. SmSPL2 Promoter Analysis in Arabidopsis

The *SmSPL2* promoter region, 1380 bp fragment upstream of the translation initiation site, was cloned and inserted into the pCAMBIA1391Z to drive the expression of *GUS*. The *Arabidopsis*-expressing *ProSmSPL2*::*GUS* was acquired via the reported method [44]. GUS histochemical staining was performed on T2 generation transgenic *Arabidopsis* at various development periods, according to a previously described protocol [45].

### 4.4. Subcellular Localization Assay of SmSPL2

To study the subcellular localization of the *Sm*SPL2, the open reading frame (ORF) of *SmSPL2* without termination codon were recombined into vector pEarleyGate103 to construct pEarleyGate103-*SmSPL2* by Gateway cloning technology [46]. The pEarleyGate103 and the recombinant vector pEarleyGate103-*SmSPL2* were transformed into onion epidermal cells, respectively, and the GFP signal was recorded, as we previously described [25].

### 4.5. SmSPL2 Transcription Activation Assays

The full-length ORF of *SmSPL2* was recombined into the pGBKT7 vector to generate recombinant vector pGBKT7-*SmSPL2* through the gateway cloning system [46]. The pGBKT7-*SmSPL2* was transformed into yeast strain AH109 using a lithium acetate mediated method [47]. The transformed cells were grown on SD/-Trp, and further cultured on SD/-Trp/-Ade/-His/X-α-gal medium to determine the transactivation activity of *Sm*SPL2, as we previously described [25].

### 4.6. Generation of Transgenic S. miltiorrhiza Plants

The miR156-resistant *SPL2* (*rSmSPL2*) was obtained through a three step PCR method (as described previously) without changing the protein sequence [48]. The primers designed for site-directed mutagenesis are listed in Appendix A. For the *SmSPL2* and *rSmSPL2*-overexpression vectors, their full-length ORFs were amplified and recombined into pEarleyGate202 to generate pEarleyGate202-*SmSPL2* and pEarleyGate202-*rSmSPL2* vectors by the gateway cloning system [46]. Transgenic *S. miltiorrhiza* plants that overexpressed *SmSPL2* or *rSmSPL2* were obtained by *Agrobacterium*-mediated transformation [43].

### 4.7. Transcriptome Analysis

Total RNAs were extracted from the roots of two-month-old *rSmSPL2*-OE2, *rSmSPL2*-OE7, and control plants using the HiPure HP Plant RNA Mini Kit (Guangzhou Meiji Biotechnology Co., Ltd., Guangzhou, China). Three plants of each line (*rSmSPL2*-OE2, *rSmSPL2*-OE7, and control) were mixed for RNA-seq by Novogene (Beijing, China). Genes with a Q value of < 0.05, a false discovery rate (FDR) < 0.001, and an absolute log_2_ fold-change (log_2_ FC) > 1.0 were identified as DEGs. We mapped all DEGs to the KEGG database terms to search for significantly overrepresented metabolic or signal transduction pathways.

### 4.8. Determination of Cytokinin and Auxin Concentrations

The fresh roots of one-month-old transgenic lines (*rSmSPL*-OE2 and *rSmSPL*-OE7) and control plants were used to calculate endogenous cytokinin and auxin concentrations by the Plant CTK/IAA ELISA Kit (mlbio, Shanghai, China).

### 4.9. HPLC Analysis of SalB and RA

To determine the SalB and RA concentrations, we collected the roots of *rSmSPL2*-OE2, *rSmSPL2*-OE7, and control plants and lyophilized the samples in a freeze-dryer (FD8-6, SIM International Group, Los Angeles, CA, USA). Phenolic acids were extracted according to a previous method [15], and determined on the UltiMate 3000 HPLC System (Thermo Fisher Scientific, Waltham, MA, USA) equipped with an Agilent ZORBAX SB-C18 column (250 mm × 4.6 mm, 5 μm). The mobile phase consisted of (A) acetonitrile–(B) 0.1% phosphoric acid with a gradient of 0–20 min, 10–20% A; 20–50 min, 20–25% A; 50–52 min, 25–10% A; and 52–53 min, 10% A. All separations were performed with a detection wavelength of 286 nm, flow rate of 1.0 mL/min, and column temperature of 28 °C.

### 4.10. Y1H Assays

The specific primers containing *Sma*I and *Sac*I sites were used to amplify *SmSPL2* and insert it into pGADT7-Rec2 to obtain pGADT7-*SmSPL2*. The pHIS2-*pTAT1*, pHIS2-*pPAL1*, and pHIS2-*p4CL9* vectors have been obtained in our previous studies [49]. The recombinant vectors were co-transferred into yeast strain Y187 using the lithium acetate mediated method [47]. The interaction between the *Sm*SPL2 and the promoter region was tested, according to the method previously reported [15].

### 4.11. Transient Transcriptional Activity (TTA) Assay in Tobacco

The full-length ORF of *SmSPL2* were inserted into the *Sma*I and *Kpn*I sites of pGreenII-62SK to obtain the effector vector pGreenII-*SmSPL2*. The *pSm4Cl9*-*LUC*, *pSmTAT1*-*LUC*, and *pSmPAL*1-*LUC* reporter vectors were obtained in our previous study [50]. The pair of effector and reporter vector were co-transferred into tobacco leaves, then monitor the transient expression of firefly luciferase (LUC) and renillia luciferase (REN), according to the previous protocol [34].

## 5. Conclusions

The overexpression of *SmSPL2* or *rSmSPL2* not only significantly reduced the accumulation of SalB but also strongly influenced root architectures, with longer lengths, fewer adventitious roots, decreased lateral root densities, and increased fresh weights. Transcriptome analysis indicated that the overexpression of *rSmSPL2* influenced root morphologies by triggering cytokinin biosynthesis and signal transduction, while inhibiting auxin biosynthesis and signaling. *Sm*SPL2 negatively regulates SalB and RA biosynthesis by directly binding to the promoters of *Sm4CL9*, *SmTAT1*, and *SmPAL1* and suppressing their expression. *SmSPL2* is a potential gene for the genetic engineering breeding of *S. miltiorrhiza*.

## Figures and Tables

**Figure 1 ijms-23-13549-f001:**
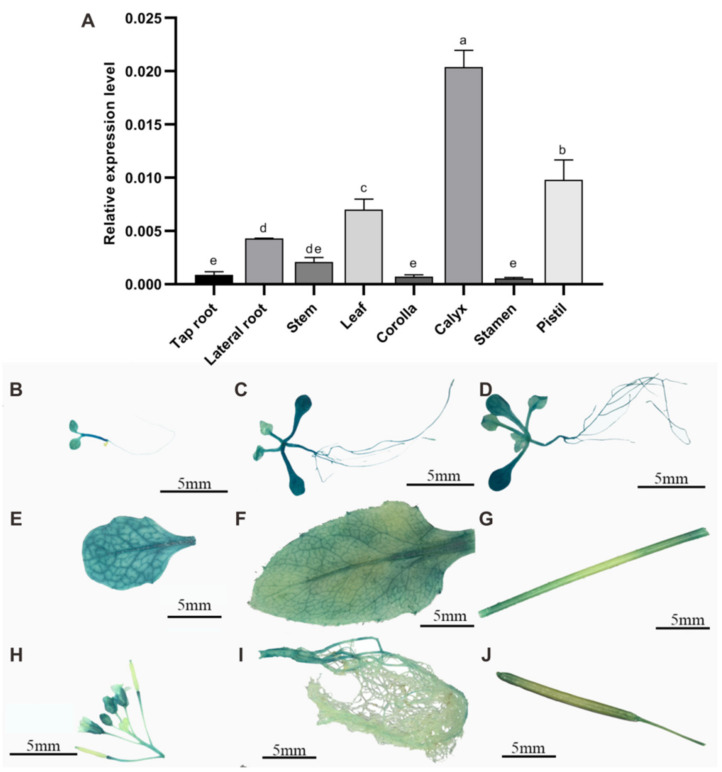
Expression profiles of *SmSPL2*. (**A**) Expression level of *SmSPL2* in different tissues. Different letters indicate significant differences between means at the *p* < 0.05 level by one-way ANOVA test (followed by Tukey’s comparisons). (**B**–**J**) GUS staining signals of transgenic Arabidopsis expressing SmSPL2pro::GUS. (**B**–**D**) Seedlings after germination at two (**B**), five (**C**), and nine (**D**) days. (**E**–**I**) Stem leaf (**E**), rosette leaf (**F**), stem (**G**), flower (**H**), and root (**I**) at the flowering stage. (**J**) Silique.

**Figure 2 ijms-23-13549-f002:**
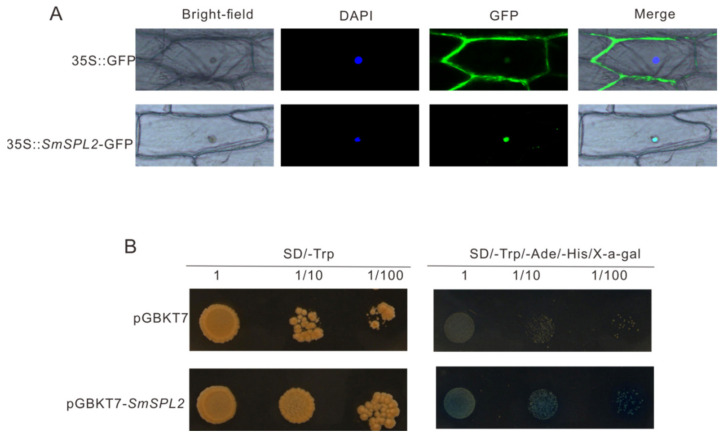
Subcellular location and transactivation activity analysis of SmSPL2 protein. (**A**) Subcellular location of SmSPL2 in onion epidermal cells. The fluorescence signal was observed via a laser scanning confocal microscope. DAPI, 4′,6-diamidino-2-phenylindole is a blue-fluorescent DNA stain used to indicate the nucleus region. (**B**) Transcriptional activation analysis of SmSPL2 protein. Yeast colonies with three different dilutions were cultured on SD/-Trp and SD/-Trp/-Ade/-His/X-α-gal media. X-α-gal: 5-Bromo-4-chloro-3-indolyl-α-D-galactoside medium, the color reaction substrate of α-galactosidase.

**Figure 3 ijms-23-13549-f003:**
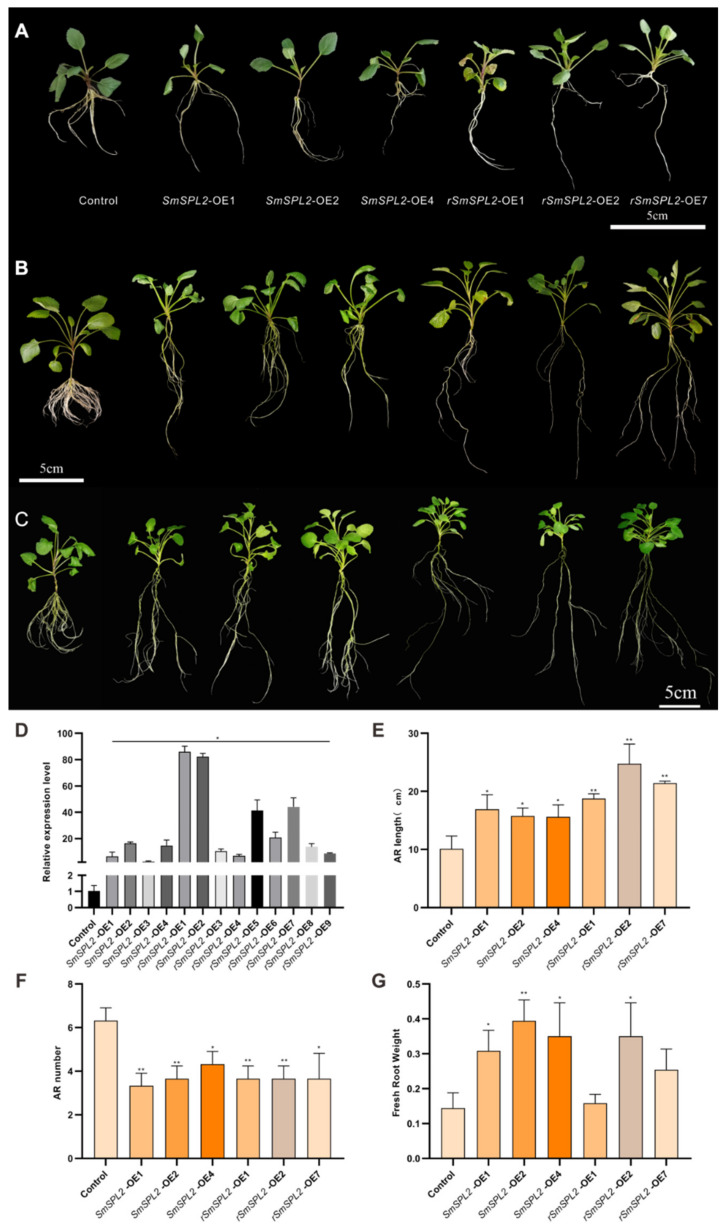
Phenotype of transgenic *S. miltiorrhiza* plantlets overexpressing SmSPL2 (SmSPL2-OE) and miR156-resistant SmSPL2 (rSmSPL2-OE). (**A**–**C**) Phenotype of 15-day-old (**A**), one-month-old (**B**), and two-month-old (**C**) transgenic *S. miltiorrhiza* plants. (**D**) Expression levels of SmSPL2 in the roots of different transgenic lines. (**E**–**G**) Adventitious root (AR) length (**E**), AR numbers (**F**), fresh weight of root (**G**) of two-month-old transgenic lines. All data are mean values of three biological replicates, with the error bar representing SD. * and ** indicate significant differences from the control at *p* < 0.05 and *p* < 0.01 level, respectively, by Student’s *t*-test.

**Figure 4 ijms-23-13549-f004:**
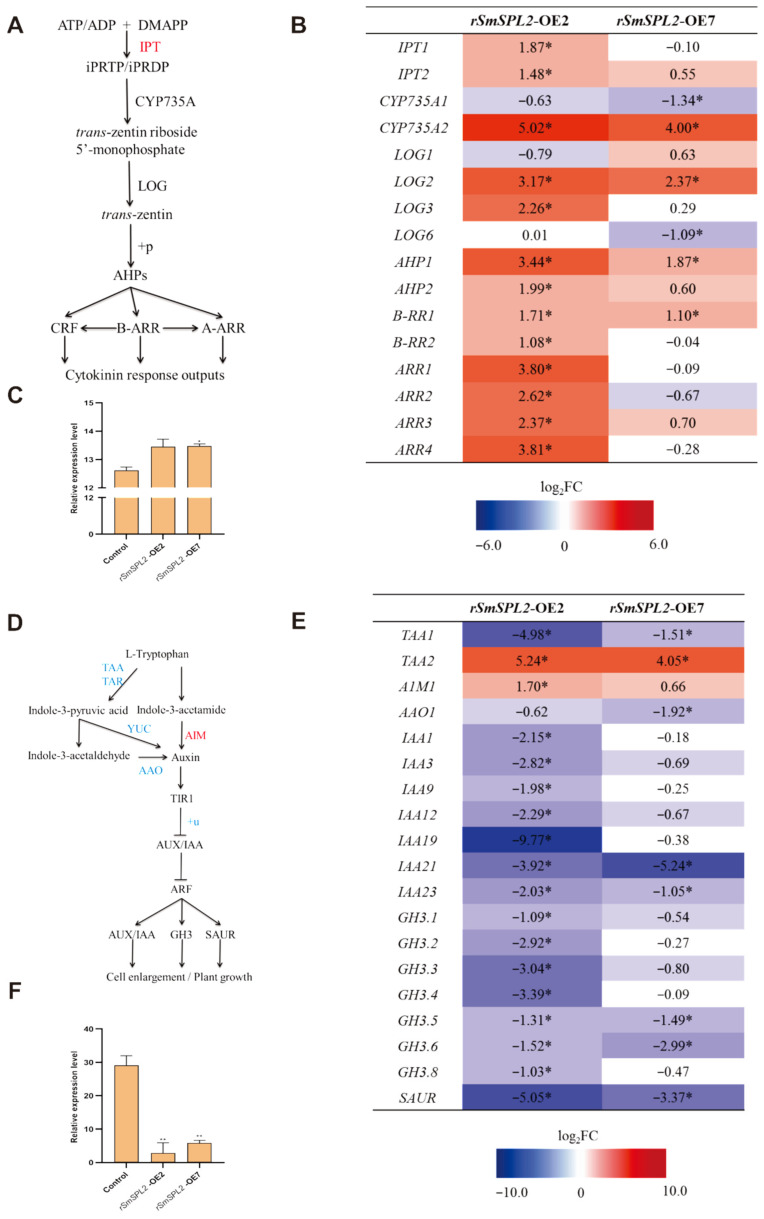
Changes in transcriptional levels of genes for cytokinin and auxin biosynthesis and signal transduction. (**A**) Proposed biosynthetic and signal transduction pathway for cytokinin. DMAPP, dimethylallyl pyrophosphate; IPT, isopentenyl transferases; CYP, cytochrome P450 enzyme; LOG, LONELY GUY; AHPs, histidine phosphotransfer proteins; CRF, cytokinin response factors; B-AAR, type-B response regulators; A-AAR, type-A response regulators. (**B**) DEGs involved in cytokinin biosynthesis and the signal transduction pathway. For each gene, the relative expression (*rSmSPL2*-OE2 vs. control and *rSmSPL2*-OE7 vs. control) was expressed in Log_2_FC. Blue boxes (gene expression is down-regulated); Red boxes (gene expression is up-regulated). * indicates differentially expressed genes. (**C**) Concentrations of cytokinin in root extracts from control and transgenic line *rSmSPL2*-OE2 and *rSmSPL2*-OE7. *, values are significantly different from control at *p* < 0.05 by Student’s *t*-test. (**D**) Proposed biosynthetic and signal transduction pathways for auxin. TAA, Tryptophan aminotransferase of Arabidopsis; TAR, Tryptophan aminotransferase related; YUC, YUCCA; AIM, Indole-3-acetamide hydrolase; AAO, Aldehyde oxidase; TIR1, Transport Inhibitor Response 1; AUX/IAA, auxin/indole acetic acid; GH3, Gretchen Hagen 3; SAUR, small auxin-up RNA. (**E**) DEGs involved in auxin biosynthesis and signal transduction pathway. For each gene, the relative expression (*rSmSPL2*-OE2 vs. control and *rSmSPL2*-OE7 vs. control) was expressed in Log_2_FC. Blue boxes (gene expression is down-regulated); Red boxes (gene expression is up-regulated). * indicates differentially expressed genes. Gene ID were listed in Appendix A. (**F**) Concentrations of auxin in root extracts from control and transgenic line *rSmSPL2*-OE2 and *rSmSPL2*-OE7. **, values are significantly different from control at *p* < 0.01 by Student’s *t*-test.

**Figure 5 ijms-23-13549-f005:**
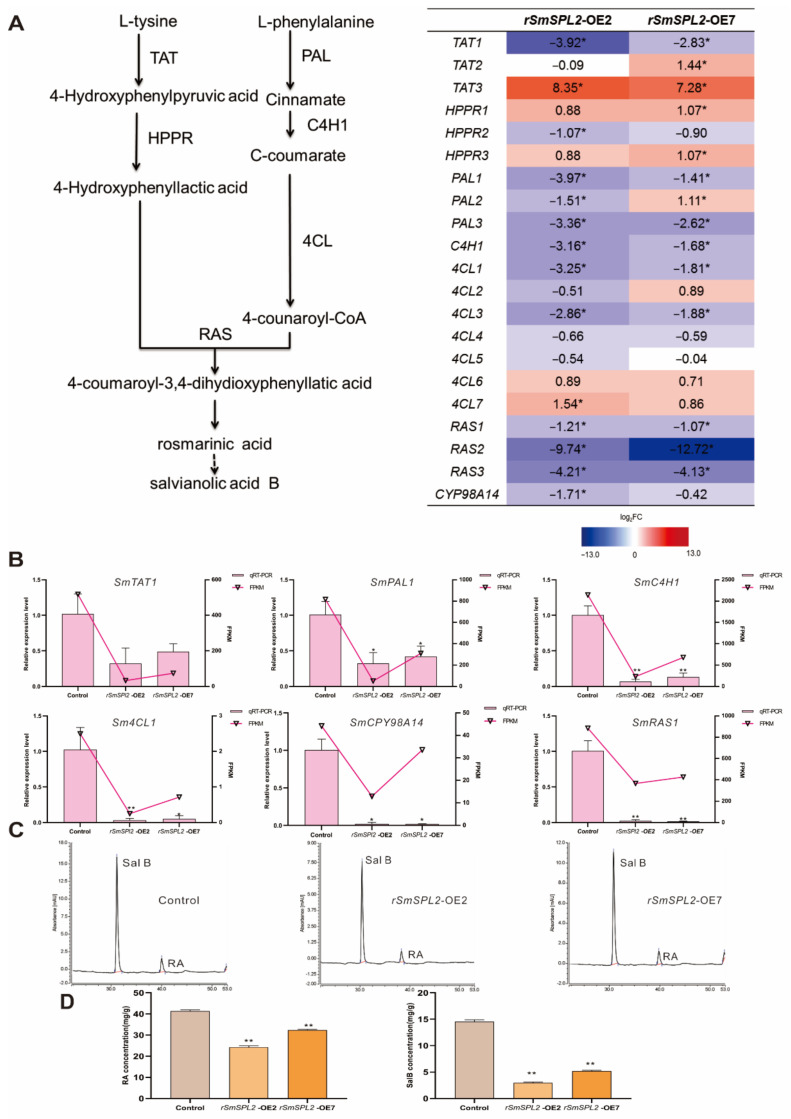
Overexpression of miR156-resistant *SmSPL2* (*rSmSPL2*) decreased the accumulation of rosmarinic acid (RA) and salvianolic acid B (SalB). (**A**) Transcriptome analysis of enzyme genes for salvianolic acid biosynthesis pathway in transgenic lines (*rSmSPL2*-OE2 and *rSmSPL2*-OE7). For each gene, the relative expression (*rSmSPL2*-OE2 vs. control and *rSmSPL2*-OE7 vs. control) was expressed as Log_2_FC. Blue boxes (gene expression is down-regulated); Red boxes (gene expression is up-regulated). TAT, tyrosine aminotransferase; HPPR, hydroxyl phenylpyruvate reductase; PAL, phenylalanine ammonia lyase; C4H, cinnamate 4-hydroxylase; 4CL, hydroxycinnamate-CoA ligase; RAS, rosmarinic acid synthase; and CYP, cytochrome P450 enzymes. (**B**) Validation by qRT-PCR of six enzyme genes involved in salvianolic acid biosynthesis in transgenic lines and control. (**C**) Chromatograms of RA and SalB in transgenic lines and the control. (**D**) RA and SalB concentrations in the roots of two-month-old transgenic lines and the control. All data are mean values of three biological replicates, with the error bar representing SD. * and ** indicate significant difference from the control at *p* < 0.05 and *p* < 0.01 level, respectively, by Student’s *t*-test.

**Figure 6 ijms-23-13549-f006:**
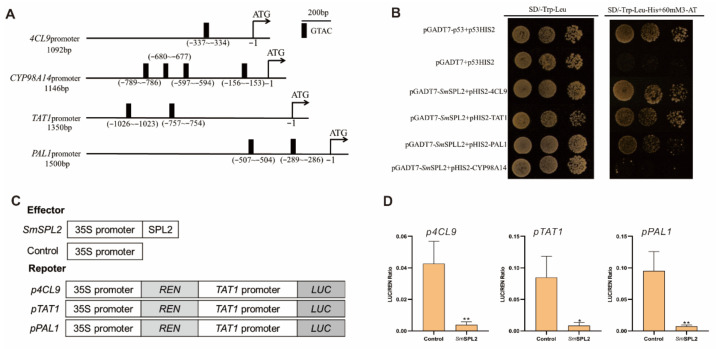
*Sm*SPL2 binds to the promoter regions of *Sm4CL9*, *SmTAT1*, and *SmPAL1* and inhibits their expression. (**A**) GTAC motifs in the promoter regions of *Sm4CL9*, *SmCYP98A14*, *SmTAT1*, and *SmPAL1*. Black rectangles represent the GTAC motif. (**B**) Yeast one-hybrid detected interactions between the *Sm*SPL6 and the promoters of *Sm4CL9*, *SmCYP98A14*, *SmTAT1*, and *SmPAL1*. The p53HIS2/pGADT7-p53 and p53HIS2/pGADT7 were the positive and negative controls, respectively. (**C**) Schematic diagram of constructs used in assays of transient transcriptional activity. (**D**) *Sm*SPL2 inhibited the expression of *Sm4CL9*, *SmTAT1*, and *SmPAL1*. Effector *Sm*SPL2 was co-transformed with reporter *p4CL9*-LUC, *pTAT1*-LUC, or *pPAL1*-LUC. All data are mean values of three biological replicates, with the error bar representing SD. * and ** indicate significant differences from the control at *p* < 0.05 and *p* < 0.01 levels, respectively, by Student’s *t*-test.

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
