# Peer review of "Transcription Factor SmSPL2 Inhibits the Accumulation of Salvianolic Acid B and Influences Root Architecture"

_ijms, 2022, doi:10.3390/ijms232113549_

Round 1

Reviewer 1 Report

In this study, Xiang-Zeng Wang et al. reported the functional analysis of SPL2 gene in model medical plant Salvia miltiorrhiza. By overexpressing either SmSPL2 or miRNA-resistant SmSPL2, authors found that transgenic lines exhibit longer length, fewer adventitious roots, decreased lateral root density. Transcriptomic analysis showed that SmSPL2 regulates important genes involved in hormone biosynthesis. Mechanistically, authors further demonstrated that several enzyme genes in the pathway of phenolic acid biosynthesis are directly suppressed by SmSPL2, which led to the reduction of medical compound salvianolic acid B in SmSPL2 OE lines. This study provided solid evidence for the biological functions of SPL2 gene in Salvia miltiorrhiza, and the experiments are performed in diversified approaches of high standard. I have no concerns regarding the current version of manuscript.  

Author Response

Thank you very much for your review.

Reviewer 2 Report

General comments for authors:

The manuscript „Transcription factor SmSpl2 inhibits the accumulation of salvianolic acid B and influences root architecture
is well written and clear in its goals and results presented. Nevertheless, I  have one suggestion.  The chapter materials and methods used in this research are not described in sufficient detail. I

Author Response

Thank you very much for your review. Since most methods used in this study have already been described in detail in our published literatures, in the chapter materials and methods we cited the reference in the chapter materials and methods, to avoid the higher duplicate rate.

Reviewer 3 Report

see attach
